# Impact of Hidradenitis Suppurativa Surgical Treatment on Health-Related Life Quality

**DOI:** 10.3390/jcm11154327

**Published:** 2022-07-26

**Authors:** Marcin Gierek, Diana Kitala, Wojciech Łabuś, Karol Szyluk, Paweł Niemiec, Gabriela Ochała-Gierek

**Affiliations:** 1Center for Burns Treatment im. Dr Sakiel, ul. Jana Pawła II 2, 41-100 Siemianowice Slaskie, Poland; dp.kitala@gmail.com (D.K.); wojciech.labus@gmail.com (W.Ł.); 2Department of Physiotherapy, Faculty of Health Sciences in Katowice, Medical University of Silesia in Katowice, 40-752 Katowice, Poland; karol.szyluk@sum.edu.pl; 3I Department of Orthopaedic and Trauma Surgery, District Hospital of Orthopaedics and Trauma Surgery, 41-940 Piekary Slaskie, Poland; 4Department of Biochemistry and Medical Genetics, Faculty of Health Sciences in Katowice, Medical University of Silesia in Katowice, 40-752 Katowice, Poland; pniemiec@sum.edu.pl; 5Dermatology Department, City Hospital in Sosnowiec, ul. Zegadłowicza 3, 41-200 Sosnowiec, Poland; g.ochala@wp.pl

**Keywords:** hidradenitis suppurativa, quality of life, surgical treatment

## Abstract

(1) Background: Hidradenitis suppurativa is a disease that affects the intimacy of patients. This disease reduces the quality of life and functioning of patients in everyday life. The surgical treatment of HS is one of the treatments for HS that can improve the quality of life. (2) Methods: The main goal of this study was to assess quality of life before the surgical treatment and after the surgical treatment of HS at Center for Burn Treatment in Siemianowice Śląskie, Poland, using the EQ-5D-5L survey before the operation and at follow-up (6 months after). (3) Results: The average quality of life measured with the EQ-5D-5L survey before therapy was 39.3 ± 20.1 (min., 0; max., 60; most frequent value, 50), whereas after surgical treatment, the mean quality of life was 89.5 ± 12.5 (min., 50; max., 100; most frequent value, 100). The average increase in the quality of life was 50.2 ± 19.5 (min., 30; max., 100; most frequent value, 30), and it was statistically significant (*p* < 0.001).

## 1. Introduction

Hidradenitis suppurativa (HS/Acne Inversa) is a chronic and painful, multifactorial inflammatory cutaneous disease of the pilosebaceous unit. In HS, normal skin is transformed into skin with abscesses formation, inflamed nodules, tunnels and scars. It mostly affects intertriginous areas of the body, such as axillae, groins, the ano-genital region, buttocks and the sub-mammary region [1,2].

The primary defect in hidradenitis suppurativa pathophysiology involves the follicular occlusion of the folliculopilosebaceous unit, followed by immune response. It is initiated by pro-inflammatory cytokines (IL-1β and TNF-α), mediators of activated T helper Th1 and Th17 cells (IFN-γ and IL-17). In addition, neutrophilic granulocytes, macrophages and plasma cells are involved in disease occurrence [3].

The prevalence of HS is unclear, ranging from 0.00033% to 4.10%; HS occurs more frequently in young adults and in women [4].

HS occurs most commonly in the third and fourth decades of life. An average delay of 7–10 years has been reported between disease onset and diagnosis. Acne inversa is associated with a number of comorbidities, e.g., metabolic syndrome, PCOS (polycystic ovary syndrome), mental health diseases such as depression, immune-mediated diseases, such as inflammatory bowel disease, spondyloarthropathy, psoriasis, alopecia areata and thyroid dysfunction [5].

Hidradenitis suppurativa is a chronic inflammatory skin disease with negative physical and psychosocial effects [6].

HS has a relevant impact on the quality of life both physically and mentally. Considering the pain, discharge, smell and associated pruritus, HS has been documented to have a negative influence on patients’ health-related quality of life [7,8,9]. Moreover, the disease has often been correlated with severe socio-economic consequences, a higher incidence of depression, the fear of stigmatization and suicide [10,11,12]. Patients with HS have a greater risk of suicidal tendencies than the general population [7,8,9].

In patients’ therapy, several methods are used, such as topical treatment, systemic antibiotic therapy, hormones, biologic therapy and surgery [13,14,15]. First-line antibiotics therapies are clindamycin and the systemic use of clindamycin and rifampicine [2]. Conservative therapy is frequently not effective enough, and surgery turns out to be the most efficient. However, adalimumab, a recombinant human TNF inhibitor, is the only approved biologic agent indicated for the treatment of moderate–severe HS, which is unresponsive or intolerant to oral antibiotics. Adalimumab has been reported to be more effective than other non-surgical treatments [16,17,18].

HS is an under-diagnosed and under-treated disease, and an interdisciplinary management approach may be needed to ensure the best outcomes in light of the associated comorbidities of the disease. Thus, HS is not only a dermatological but also a gynecological and surgical disease [19]. The mainstay of traditional surgery is a wide local excision, and it can result in a disease-free state where the excision is performed. Only radical surgical excision can securely prevent recurrence. After radical excision, wound healing by secondary intention as well as the use of split skin grafts have been described to be effective [19]. Excision can typically be limited to a superficial subcutaneous tissue, with deeper excision being based on visible disease extension [20,21].

The aim of this study was to assess health-related life quality before the surgical treatment and after the surgical treatment of HS at Center for Burn Treatment in Siemianowice Śląskie, Poland, using the EQ-5D-5L survey.

## 2. Materials and Methods

### 2.1. Data Gathering

On the basis of patients’ medical records from 2020 to 2021 of every subsequent patient with recognition of HS and with surgical treatment at Center For Burn Treatment in Siemianowice Śląskie, Poland, data were collected. We included in the research study 21 patients with HS, each one with surgical treatment and 2-week and 6-month post-op follow-ups. Inclusion criteria: diagnosed HS for at least 1 year, females, males, Hurley I–III score and patients with comorbidities (such as diabetes, hypertension, obesity, metabolic disease, nicotinism). We included only patients who gave their consent to participate in the study. Criteria for exclusion from the study: patients who did not consent to participate in the study, cancer, pregnancy and women during breastfeeding.

We collected basic characteristics of patients, such as age, sex, height, weight, BMI, comorbidities (diabetes, hypertension, metabolic disease, obesity), the severity of the disease (Hurley scale), location of the lesions, the number of years from the first symptoms and the number of years from the first symptoms to the correct diagnosis of the disease. We collected data on the length of hospitalization (days) and surgical methods used in our Center. Data about patients and surgical treatment were gathered in Excel (Microsoft Corp., Redmond, WA, USA) on the basis of Medicus Hospital Informatics System (Gabos Software Sp. Z o. o., Tarnowskie Góry, Poland). The study was conducted in line with the Helsinki Congress (revised in 2013) and the Istanbul Declaration. All patients agreed to complete the questionnaire and consented to the participation of medical data from their treatment in this study. The institutional ethical review board of Center For Burn Treatment, Siemianowice Śląskie, Poland, approved the research study (No. 1/2022/CLO). The authors declare no conflict of interest.

### 2.2. Quality-of-Life Survey

Each patient completed a questionnaire on the health-related life quality (Polish version of EQ-5D-5L) before surgical intervention and after (on the follow-up visits). The EQ-5D-5L survey is a standardized document, and it did not require any validation. The survey consisted of five questions. The questions concerned areas of everyday life such as mobility, self-service, ordinary activities (e.g., work, study, etc.), pain/discomfort and anxiety/depression. For each question, five graded answers were prepared—starting with the answer stating that the patient has no problems with the question asked and ending with answers regarding the inability to perform a given activity. Additionally, the questionnaire included the health-related life-quality scale. The scale ranged from 0 to 100, where 0 was the worst health imaginable and 100 was the best health imaginable. Patients completed a new questionnaire before the surgical treatment and postoperatively after 6 months. The study included patients with HS who were treated surgically at Center For Burn Treatment in Siemianowice Śląskie, Poland, in the years 2020–2021. Patients who did not agree to the health-related life-quality questionnaire before and after surgery were excluded from the study.

### 2.3. Statistical Analysis

All tests were performed with the use of STATISTICA 13.0 software (TIBCO Software Inc., Tulsa, CA, USA). The normality of the distribution was analyzed with the Shapiro–Wilk test. Since all quantitative variables had non-normal distribution, the results were presented as medians with their quartile deviations (±QD). Only non-parametric tests were used for their analyses, namely, the Mann–Whitney U test (for dichotomous grouping variables) and the Kruskal–Wallis test (for grouping variables with more than two classes). The differences in the quantitative data of matched samples were computed using the Wilcoxon signed-rank test. Spearman’s rank correlation coefficient (r_s_) was used to interpret the results of the correlation analyses. The *χ*^2^ test was used for the analyses of the qualitative data. Fisher’s exact was applied when the size of any group was less than 10. The significance level was set at *p* < 0.05. For multiple comparisons, *p*-values were corrected using the Bonferroni correction. A power analysis was not performed due to the small sample size, which is a result of the rarity of HS.

## 3. Results

### 3.1. General Data

General characteristics of the hidradenitis suppurativa patients are in Table 1.

### 3.2. Types of Surgical Interventions

We would like to present examples of surgical reconstructive techniques that the patients underwent in our hospital (Figure 1).

### 3.3. Health-Related Life Quality

We noticed a significant improvement in the quality of life of patients with HS at the six-month follow-up relative to the condition before surgery to remove HS (median ± QD: 50.00 ± 10.00 vs. 90.00 ± 9.50, respectively, *p* < 0.000). The statistical significance of the differences was observed in HS patients in Hurley stage I (*p* = 0.043) and in Hurley stage III (*p* = 0.001) but not in Hurley stage II (*p* = 0.109), probably due to the small sample size. In the case of individual components of the EQ-5D-5L questionnaire, there was a significant improvement in subjectively declared problems with walking, self-care, regular activities, pain, anxiety and depression (*p* < 0.001, for each component).

We analyzed the demographic and clinical factors that could potentially affect the quality of life of HS patients. Age did not correlate with quality of life but with the time from first symptoms to final diagnosis (r_s_ = 0.67, *p* < 0.050) and the duration of the disease (r_s_ = 0.69, *p* < 0.050). Before the procedure, women and men had similar levels of declared health-related life quality (median ± QD: 50.00 ± 15.00 vs. 50.00 ± 20.00, respectively) and these differences were not statistically significant (*p* = 0.801). At the half-year observation, men were characterized by a higher quality of life than women (median ± QD: 100.00 ± 7.50 vs. 90.00 ± 9.50, respectively), although also in this case, the differences did not reach statistical significance (*p* = 0.075). This does not change the fact that the improvement in the quality of life observed after the procedure was significant both in women (*p* = 0.003) and in men (*p* = 0.005).

The severity of the disease is a factor that may have a potential impact on the health-related life quality of patients with HS. In the current study, we showed that the preoperative quality of life was significantly lower in patients in Hurley stage I than in those in Hurley stage II (*p* = 0.031). The lack of association between the quality of life and the Hurley HS scale in the postoperative period suggests that intervention increased the quality of life regardless of the severity of the disease. The detailed data presented in Table 2 additionally list the factors differentiating patients in different stages of the disease.

Factors such as the duration of disease, time from first symptoms to final diagnosis and time of hospitalization did not significantly (*p* > 0.050) improve the health-related life quality of patients with HS (data not shown). In the case of surgical intervention, patients with rotatory flaps were characterized by a higher quality of life than those with other types of surgical intervention (median ± QD: 95.00 ± 7.50 vs. 75.00 ± 15.00, respectively); however, also in this case, no statistical significance was demonstrated (*p* = 0.079).

We also checked with which parameters the number of points obtained in the quality-of-life questionnaire correlated, both for the entire study group and for patients in individual Hurley stages (Table 3). It was shown that before the procedure, the number of points obtained in the EQ-5D-5L questionnaire correlated with the time elapsed from the onset of the first symptoms to the diagnosis of HS. Before surgery, the health-related life quality of HS patients in Hurley stage I positively correlated with age and negatively correlated with the time from the first symptoms to the diagnosis of HS. After surgery in this stage, a negative correlation of QoL with hospitalization and a positive one with duration of disease were also observed. In the case of HS patients in Hurley stages II and III, no correlations were found between the health-related life quality and the quantitative variables studied (data not shown).

## 4. Discussion

Hidradenitis suppurativa is a chronic disease that affects many aspects of life. Many years of ineffective treatment pose a great challenge for patients and for physicians treating HS. Our study concluded a significant improvement in the health-related life quality of patients with HS at the six-month follow-up, relative to the condition before surgery (median ± QD: 50.00 ± 10.00 vs. 90.00 ± 9.50). The statistical significance of the differences was observed in the group of HS patients in Hurley stage I (*p* = 0.043) and in Hurley stage III (*p* = 0.001). We found that factors such as the duration of disease and time from first symptoms to final diagnosis did not significantly (*p* > 0.050) improve the life quality of patients with HS. Our study proved that surgical treatment improved the health-related life quality of patients with hidradenitis suppurativa.

The importance of measuring the health-related life quality of patients with HS has been described in numerous studies [22,23,24]. There is no doubt that HS is a chronic disease with a large impact on the patient’s life due to its chronicity, recurrence, impact on body image, sexual health, pain and odor, among other impacts [20]. Due to the characteristics of the disease, often embarrassing symptoms, patients feel stigmatized, and HS frequently leads to depression and other psychiatric symptoms and severe socio-economic problems. Sometimes patients commit suicide [9,10,12].

Data concerning quality-of-life issues (QoL) after HS surgery are very limited. The largest study amongst those recently published assessed 149 patients in Hurley stages I–III [20]. Grimstad et al. compared surgical treatment, mainly CO2 laser procedures and deroofing. In Grimstad et al., in the CO_2_ laser group, changes from baseline in Sartorius, NRS severity and QoL increase scores were significant (all *p* < 0.001) [25]. Grimstad et al. also compared medical treatments and surgical treatment, and both methods (surgical and medical treatment, i.e., topical treatment). Grimstad et al. showed that surgical treatment was more effective than the non-biologic medical treatments. The study provided limited evidence for the combination of medical and surgical therapies in patients with HS [25].

The quality-of-life assessment of surgical treatment by Gibrila et al. compared two surgical techniques: artificial skin and perforator flaps. A total of 47 patients were included in Gibrila et al. study between January 2015 and September 2017, including 27 patients in the artificial dermal group and 20 patients in the perforator flap group. The study showed a significant increase in the quality of life in both groups (*p* < 0.05), and this was higher in the perforator-flap group (*p* < 0.001) [26].

In other study, the QoL after complex wound closure was measured in 27 patients who underwent pedicled-perforator-flap reconstruction after wide local excision (WLE). The mean DLQI score significantly improved after six months, from 21.3 ± 4.8 to 5.0 ± 3.0 at the last follow-up (*p* < 0.0001). Furthermore, DLQI scores were not influenced by complications such as having to have a further operation [26].

Marchesi et al. proved that wide local excision followed by pedicled-perforator-flap reconstruction allowed the radical excision of HS areas with short postoperative healing periods. Dermatology Quality of Life Index scores confirmed the high levels of patients’ satisfaction. The disadvantages of this technique include a difficult learning curve, long operating time and a nonnegligible complication rate [27].

Our study showed results similar to those obtained by other authors. We confirmed that the health-related life quality was significantly increased after surgical treatment. The average health-related life quality measured with the EQ-5D-5L survey before therapy was 39.3 ± 20.1, whereas after surgical treatment, the mean health-related life quality was 89.5 ± 12.5.

The limitations of this manuscript may seem to include the small number of participants (21 patients). However, it should be emphasized that there are a very large number of patients who are not properly diagnosed, which may be evidenced by the long period from the appearance of the first symptoms to the correct diagnosis of HS. This work confirmed the effectiveness of the surgical treatment of HS, which can be an ideal complement to the available therapies (biological and non-biological). Work on the quality of life and health-related life quality in patients with HS should be continued, especially considering the psychological aspects in these patients. The surgical treatment of HS improves the health-related life quality and thus the overall quality of life in patients with HS.

## 5. Conclusions

HS is a challenging disease that is difficult for the patient and the attending physician. Systemic treatment is often ineffective and is associated with a very high patient dissatisfaction. In this study, we prove at a 6-month follow-up after surgery that surgical treatment is effective, satisfies patients and significantly improves their quality of life. HS is a chronic disease that can reduce the health-related quality of life. Surgical treatment effectively improves the postsurgical health-related quality of life.

## Figures and Tables

**Figure 1 jcm-11-04327-f001:**
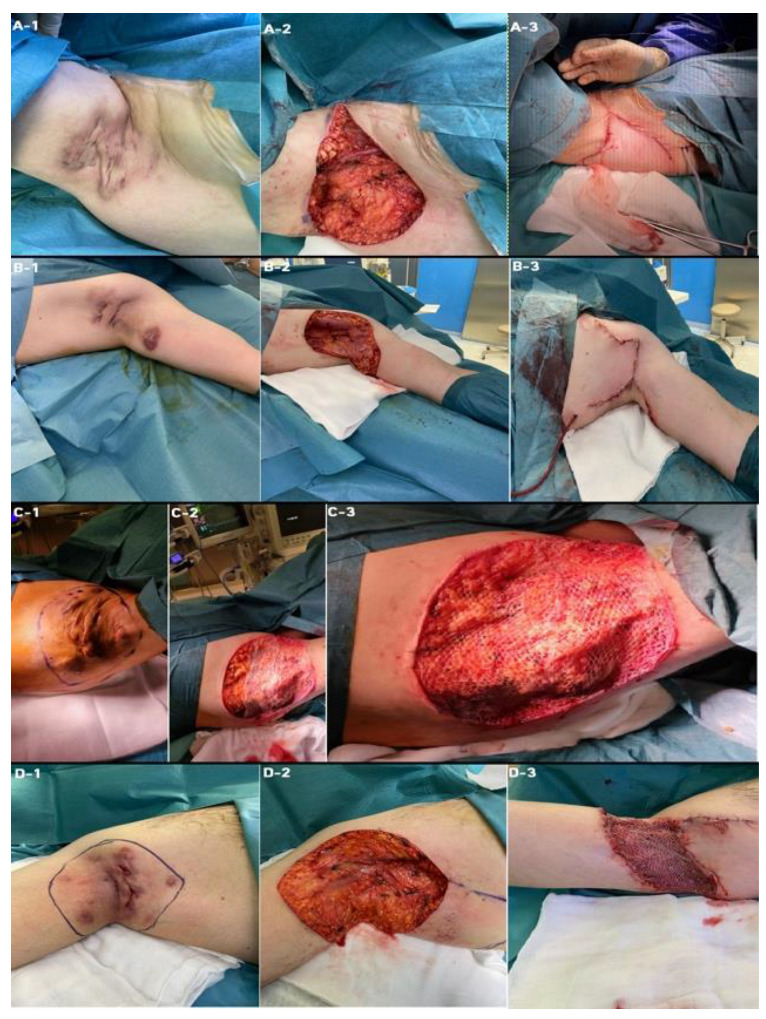
Types of surgical reconstructive techniques used in the surgical treatment of HS. (**A**) Flap reconstruction: left—HS of right axilla (**A-1**); center—wide excision (**A-2**); right—rotatory-flap reconstruction (**A-3**). (**B**) Flap reconstruction: left—HS of left axilla (**B-1**); center—wide excision (**B-2**); right—bilobed reconstruction (**B-3**). (**C**) Split-thickness skin graft: left—HS of left axilla and chest (**C-1**); center—wide excision and split-thickness skin graft (**C-2**); right—reconstruction with split-thickness skin graft (**C-3**). (**D**) Mixed technique (Flap + STSG): left—HS of right axilla (**D-1**); center—wide excision (**D-2**); right—reconstruction with a rotatory flap and split-thickness skin graft (**D-3**).

**Table 1 jcm-11-04327-t001:** General and clinical characteristics of the study group.

Characteristic		
General	n (%)	21 (100.00)
	Age, median (±QD)	38.00 (10.00)
	Males, n (%)	10 (47.61)
	Females, n (%)	11 (52.38)
	BMI, median (±QD)	33.00 (2.85)
HS	Hurley stage of disease, n (%)	
	- I	5 (23.80)
	- II	3 (14.28)
	- III	13 (61.90)
	iHS4 score	
	- Mild (≤3 points)	5 (23.80)
	- Moderate (4–10 points)	3 (14.28)
	- Severe (≥11 points)	13 (61.90)
	Location of HS, n (%)	
	- armpit	15 (71.43)
	- groin	3 (14.29)
	- buttocks	1 (4.76)
	- vulva	1 (4.76)
	- scrotum	1 (4.76)
	Time from first symptoms to final diagnosis in years, median (± QD)	8.00 (5.00)
	Duration of disease in years, median (± QD)	10.00 (4.50)
Intervention	Type of intervention, n (%)	
	- rotatory flap	17 (80.95)
	- STSG	2 (9.52)
	- rotatory flap + STSG	1 (4.76)
	- debridement + VAC + STSG	1 (4.76)
	Time of hospitalization in days, median (± QD)	5.00 (2.00)
6-month follow-up	Wound healed completely, n (%)	17 (80.95)
	Keloid, n (%)	2 (9.52)
	Wound healed completely with HS recurrence, n (%)	1 (4.76)
	Wound during healing process, n (%)	1 (4.76)
Comorbidities	Diabetes mellitus	3 (14.29)
	Metabolic syndrome	4 (19.05)
	Hypertension	9 (42.86)

Legend: BMI—body mass index; HS—hidradenitis suppurativa; STSG—split-thickness skin graft; QD—quartile deviation; VAC—vacuum-assisted closure.

**Table 2 jcm-11-04327-t002:** Clinical characteristics of HS patients in regard to the Hurley stage of Hidradenitis suppurativa.

Characteristic	Median ± QD			*p*-Value			
	Hurley Stage of HS			Kruskal–Wallis	I vs. II	I vs. III	II vs. III
	I	II	III				
EQ-5D-5L before, points	40.00 ± 20.00	60.00 ± 0.00	50.00 ± 10.00	0.024	0.031	1.000	0.059
EQ-5D-5L after, points	100.00 ± 10.00	100.00 ± 0.50	90.00 ± 5.00	0.208	-	-	-
BMI	26.60 ± 4.35	33.50 ± 1.15	30.80 ± 1.80	0.085	-	-	-
Age, years	22.00 ± 0.50	45.00 ± 4.50	42.00 ± 2.00	0.033	0.068	0.076	1.000
Time from first symptoms to final diagnosis, years	1.00 ± 0.50	15.00 ± 2.00	12.00 ± 4.00	0.001	0.001	0.014	0.296
Time of hospitalization, days	4.00 ± 1.00	4.00 ± 1.00	7.00 ± 1.50	0.014	1.000	0.023	0.306
Duration of disease, years	4.00 ± 1.50	16.00 ± 2.50	14.00 ± 4.50	0.005	0.011	0.026	0.737

Legend: HS, hidradenitis suppurativa, BMI, body mass index; EQ-5D-5L, questionnaire of health-related life quality; QD, quartile deviation.

**Table 3 jcm-11-04327-t003:** Spearman’s rank correlation coefficients (r_s_) between factors potentially influencing health-related life quality.

Variables	EQ-5D-5L Before	EQ-5D-5L After	BMI	Age	From First Symptoms to Final Diagnosis	Time of Hospitalization	Duration of Disease
All patients							
EQ-5D-5L before	-	0.31	0.30	0.41	0.55 *	0.05	0.43
EQ-5D-5L after	0.31	-	0.10	−0.21	0.03	−0.42	−0.05
Stage I patients							
EQ-5D-5L before	-	−0.44	−0.63	0.89 *	−0.89 *	0.66	−0.79
EQ-5D-5L after	−0.44	-	0.59	−0.67	0.67	−0.89 *	0.89 *

* *p* < 0.050. Legend: BMI, body mass index; EQ-5D-5L, questionnaire of health-related life quality.

## Data Availability

Not applicable.

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
