# Peer review of "Impact of Hidradenitis Suppurativa Surgical Treatment on Health-Related Life Quality"

_jcm, 2022, doi:10.3390/jcm11154327_

Round 1

Reviewer 1 Report

In this paper, Gierek et al. showed that surgical treatment improves the QOL of hidradenitis suppurativa (HS) patients using EQ-5D-5L survey. Although some similar articles have already been published, this paper is well-written and should be informative for all dermatologists. My minor comments are as follows:

1.      Adalimumab is now widely used in HS patients as a conservative treatment and has been reported to be more effective than other non-surgical treatments such as topical steroids or oral antibiotics, etc. Please shortly mention this point in the Introduction or Discussion section.

2.      Please spell out PCOS.

3.      I couldn’t understand what the authors want to say in the last sentence of Conclusion section. Please check the sentence.

Author Response

We are very grateful for all valuable suggestions, comments and remarks of the Reviewer 1.

Any changes to the manuscript are highlighted in yellow in resubmitted form of this manuscript.

We have expanded the literature on the effectiveness of Adalimumab according to the Reviewer's suggestions. 

Thank you very much for noticing that we didn't explain PCOS. It is corrected in current form of this manuscript.

The last sentence in Conclussions has been corrected to be more understandable, thank you very much for this important comment.

We appreciate the all sugesstions of Reviewer 1, which introduces a significant new value to this work. We would like to thank you for all valuable comments that will significantly improve the quality of this manuscript. 

Sincerely yours,

Marcin Gierek and Karol Szyluk

Reviewer 2 Report

The study by Gierek et al. aims to evaluate the QoL in HS patients before and after surgical treatment. The paper is well written, however, there are some observations that need to be addressed. Please note the following comments:

1.     In this study, the health-related quality of life by EQ-5D-5L questionnaire has been evaluated in HS patients before and after surgical treatment. The term "health-related life quality" should be reported in the text and title; the concept of “QoL” should involve also other characteristics and variables which have not been considered by the authors. The perceived influence of HS on QoL should be evaluated also by DLQI scores (doi: 10.3390/life11010034).

2.     In the introduction section should be considered also recent publications on HS (doi: 10.1080/17512433.2020.1762571; doi: 10.3390/cells10082094).

3.     The number of the study population is quite small, and the number of patients is not well balanced in the analyzed subgroups (5 mild HS, 3 moderate HS, and 13 severe HS), so the statistical significance cannot be achieved for some of the comparisons between the groups. This is a limitation of the study that should be reported in the manuscript. Have you applied any statistical tests for the sample size calculation? The multicollinearity test should be performed to evaluate the collinearity among the analyzed variables.

4.     The entry/exclusion criteria for patients’ enrolment should be reported in material and method section. Ethical approval for this study should be also reported.

5.     The iHS4 score should be also included in the table describing patients´ clinical characteristics.

Author Response

We are very grateful for all valuable suggestions, comments and remarks of the Reviewer 2.

Any changes to the manuscript are highlighted in yellow in resubmitted form of this manuscript.

We have expanded literature references (6 manuscripts) – all changes are highlighted in yellow in reference section.

Our answer to comment 1:

We would like to thank you very much for this significant attention. Reviewer 2 is absolutely right - our questionnaire is not a full Quality Of Life questionnaire but health-related life quality. As suggested by the Reviewer, we changed the title and in the text quality of life to health-related life quality. Reviewer 2 is right that the quality of life should also be supplemented with the DLQI questionnaire. In our work, we used the EQ-5D-5L questionnaire, however, we would like to add that our work is prospective and when the group of patients studied increases, we will definitely use the DLQI questionnaire in subsequent publications. Thank you very much for the significant comments that have definitely improved the quality of this work

Our answer to comment 2 :

Reviewer 2 is absolutely right, the work required improvement with further references. We expanded the references with 6 new items and included the publications suggested by the reviewer 2 ( doi: 10.1080/17512433.2020.1762571; doi: 10.3390/cells10082094). Thank you very much for your invaluable help, these publications will significantly improve the quality of this work.

Our answer to comment 3 :

We would like to thank to Reviewer 2 for this valuable suggestion.

A power analysis was not performed due to the small sample size which is a result of the rarity of HS. When comparing patients with particular stages of HS, we decided to use the simplest quantitative analysis test (Kruskal Wallis test), the results of which were verified by post hoc analysis . The Wilcoxon signed-rank test was used to the analysis of QoL (to compare the results before and after surgical intervention). Due to the small study group, we did not perform a multivariate analysis, which would not give reliable results for a matrix of 21 patients. On the other hand, the aim of our current work was to evaluate the impact of surgery on the quality of life of patients with HS. Factors that differentiate patients with different stages of the disease and potentially affect their quality of life were additionally analyzed. Please note that we did not show statistically significant differences in the quality of life of patients of particular stages of HS in the postoperative follow-up, but these differences were visible before surgery, which could be due to factors differentiating patients with particular stages of HS (like time from first symptoms to final diagnosis, duration of disease).  The differences in the size of the groups of patients with different stages of the disease result from such a distribution (mainly severe cases decide on the procedure), so we had no influence on this aspect of the study.

We mentioned the limitations of study group in discussion section and, also in the material methods – both fragments are highlighted in yellow in the manuscript.

We are very grateful for this comment, because we will use suggested methods in future publications, when our study group will increase. Nowadays, HS treated surgically is still rare case, and our Center is in our country one of 3 Surgical Departments which treats hidradenitis suppurativa. Still is the problem with correct diagnosis, but it seems to be a general worlds problem in HS treatment. We will consider your helpful suggestions in future manuscripts. Thank you very much for these suggestions.

Our Answer to Comment 4 :

Thank you very much for this valuable suggestion and comment. According to this suggestion we add inclusion and exclusion criteria in Material and Methods section. Ethical approval is also included in the text. All new content is highlighted in yellow. We appreciate these important suggestions, there is no doubt that all of your suggestions will improve quality of this manuscript.

Our answer to Comment 5 :

We are very grateful for noticing this information. We have already included iHS4 score in patients characteristics table.

We would like to thank you for all important, valuable comments, suggestions, opinions. All of suggestions will improved this manuscript and the quality of this study. Thank you very much!

Sincerely yours,

Marcin Gierek and Karol Szyluk

Round 2

Reviewer 2 Report

The authors improved the manuscript as suggested. So, in my opinion, the manuscript should be accepted.